Acute hypothalamic suppression significantly affects trabecular bone but not cortical bone following recovery and ovariectomy surgery in a rat model

Yingling Vanessa R. 1 2 3 vanessa.yingling@csueastbay.edu
Mitchell Kathryn A. 2
Lunny Megan 2
1 Department of Kinesiology, California State University, East Bay , Hayward, CA , United States
2 Department of Kinesiology, Temple University , Philadelphia, PA , United States
3 Department of Anatomy and Cell Biology, Temple University , Philadelphia, PA , United States
Leonhard-Marek Sabine
Electronic publication date: 2016 Jan 12
Publication date: 2016
Volume: 4
Electronic Location ID: e1575
Received 2015 Sep 11; Accepted 2015 Dec 16
Copyright: ©2016 Yingling et al.
Copyright year: 2016
Copyright holder: Yingling et al.
License: This is an open access article distributed under the terms of the Creative Commons Attribution License, which permits unrestricted use, distribution, reproduction and adaptation in any medium and for any purpose provided that it is properly attributed. For attribution, the original author(s), title, publication source (PeerJ) and either DOI or URL of the article must be cited.
License URL: https://creativecommons.org/licenses/by/4.0/

Keywords: Bone strength, Delayed puberty, OVX, GnRH antagonist, Bone adaptation

Funding: National Institutes of Health R03AR057518 Research reported in this publication was supported by the National Institute of Arthritis and Musculoskeletal and Skin Diseases, part of the National Institutes of Health, under Award Number R03AR057518. The content is solely the responsibility of the authors and does not necessarily represent the official views of the National Institutes of Health. The funders had no role in study design, data collection and analysis, decision to publish, or preparation of the manuscript.

==============================
Background. Osteoporosis is “a pediatric disease with geriatric consequences.” Bone morphology and tissue quality co-adapt during ontogeny for sufficient bone stiffness. Altered bone morphology from hypothalamic amenorrhea, a risk factor for low bone mass in women, may affect bone strength later in life. Our purpose was to determine if altered morphology following hypothalamic suppression during development affects cortical bone strength and trabecular bone volume (BV/TV) at maturity.

Methods. Female rats (25 days old) were assigned to a control (C) group (n = 45) that received saline injections (.2 cc) or an experimental group (GnRH-a) (n = 45) that received gonadotropin releasing hormone antagonist injections (.24 mg per dose) for 25 days. Fifteen animals from each group were sacrificed immediately after the injection protocol at Day 50 (C, GnRH-a). The remaining animals recovered for 135 days and a subset of each group was sacrificed at Day 185 ((C-R) (n = 15) and (G-R) (n = 15)). The remaining animals had an ovariectomy surgery (OVX) at 185 days of age and were sacrificed 40 days later (C-OVX) (n = 15) and (G-OVX) (n = 15). After sacrifice femurs were mechanically tested and scanned using micro CT. Serum C-terminal telopeptides (CTX) and insulin-like growth factor 1 (IGF-1) were measured. Two-way ANOVA (2 groups (GnRH-a and Control) X 3 time points (Injection Protocol, Recovery, post-OVX)) was computed.

Results. GnRH-a injections suppressed uterine weights (72%) and increased CTX levels by 59%. Bone stiffness was greater in the GnRH-a groups compared to C. Ash content and cortical bone area were similar between groups at all time points. Polar moment of inertia, a measure of bone architecture, was 15% larger in the GnRH-a group and remained larger than C (19%) following recovery. Both the polar moment of inertia and cortical area increased linearly with the increases in body weight. Following the injection protocol, trabecular BV/TV was 31% lower in the GnRH-a group compared to C, a similar deficit in BV/TV was also measured following recovery and post-OVX. The trabecular number and thickness were lower in the GnRH-a group compared to control.

Conclusion. These data suggest that following a transient delay in pubertal onset, trabecular bone volume was significantly lower and no restoration of bone volume occurred following recovery or post-OVX surgery. However, cortical bone strength was maintained through architectural adaptations in the cortical bone envelope. An increase in the polar moment of inertia offset increased bone resorption. The current data are the first to suppress trabecular bone during growth, and then add an OVX protocol at maturity. Trabecular bone and cortical bone differed in their response to hypothalamic suppression during development; trabecular bone was more sensitive to the negative effects of hypothalamic suppression.

Introduction

Osteoporosis is “a pediatric disease with geriatric consequences” (Golden, 2000). Bone morphology and tissue quality co-adapt during ontogeny for sufficient bone strength (Jepsen, 2011). Suboptimal bone strength in individuals who do not reach peak bone mass during childhood or adolescence may contribute to the development of fractures later in life (Bachrach, 2001). Therefore, factors during the developmental years such as, hypothalamic amenorrhea, a risk factor for low bone mass in women, may affect bone strength later in life.

The timing of puberty has emerged as a crucial factor in bone strength development. Peak bone mineral accrual rate occurs at puberty (Warren & Stiehl, 1999), with an accrual of 26% of adult total bone mineral within 2 years of pubertal onset (Pitukcheewanont et al., 2013). However, a delay in the timing of puberty is one factor among many that correlate with low bone mass in young women (Nattiv et al., 1994; Wiggins & Wiggins, 1997; Yingling & Khaneja, 2006). Warren et al. (2002) found the positive correlation between the age of menarche and stress fracture occurrence to be stronger than the age of menarche and bone mineral density (BMD) (Warren et al., 2002). Yet the effect of a delayed puberty on peak bone strength at maturity particularly post-menopause when estrogen levels in women decrease and bone mass is lost remains unclear. Young women with delayed pubertal onset may not build a strong skeleton and be at risk for fracture in the short term and later in life, in particular post-menopause.

Multiple factors affect the structural development of the skeleton; in particular estrogen levels during growth are an important factor in the pathogenesis of bone fragility (Drinkwater et al., 1984). The delay of menarche and infrequent menstrual cycles decrease estrogen levels during adolescence and decrease peak bone mass (Myerson et al., 1992; Warren et al., 2003; Nattiv et al., 2007). Investigators have identified bone densities in young (17–35 yr) athletic women with decreased estrogen levels similar to the bone densities of 51-year-old women (Myerson et al., 1992; De Souza et al., 2008). On the structural level, cortical width is established during puberty in females by endosteal apposition. Low estrogen levels during growth may result in a thinner cortex if the increased endosteal resorption (De Crée, 1998) is not offset by an increased periosteal apposition (Loucks, Verdun & Heath, 1998). The relative cellular activity on these bone surfaces affects bone size, a critical element of bone strength. In trabecular bone, post-menopausal women have significantly lower bone volume and less trabeculae (Akhter et al., 2007). Trabecular number and thickness are decreased following ovariectomy surgery, a model of post-menopausal bone loss, in both mature and young animals (Wronski, Dann & Horner, 1989; Wronski et al., 1989; Sims et al., 1996; Roux et al., 1996; Lane et al., 1998; Bourrin et al., 2002). Animal models of low estrogen (hypothalamic suppression) during adolescence report deficits in cortical bone strength (Yingling & Khaneja, 2006; Yingling & Taylor, 2008) and trabecular structure (Rakover et al., 2000; Yingling et al., 2007), and therefore it is important to determine if these deficits remain and whether they are exacerbated during the menopause. Specifically, the long term effects of hypothalamic suppression during development at a time point at maturity, in particular following estrogen loss due to the menopause modeled by ovariectomy surgery needs to be investigated.

The purpose of this study was to investigate the effect of a delay in puberty on bone strength and structure immediately post-puberty and at maturity in female rats. This model offers an opportunity to reproduce an environment of delayed puberty and to investigate the effect on bone strength and structure at critical time points throughout the life span. The investigative hypothesis predicts that administration of a GnRH antagonist prior to the onset of the first estrus cycle would suppress the increase in estrogen levels associated with the onset of puberty and impede the development of cortical and or trabecular bone strength and structure acutely and at maturity.

Methods

Animal protocol

Female rats (25 days) (Charles Rivers Laboratories, Wilmington, MA, USA) were assigned to a control (C) group (n = 45) that received saline injections (.2 cc) for 25 days or an experimental group (GnRH-a) (n = 45) that received gonadotropin releasing hormone antagonist injections (.24 mg per dose) (Luteinizing Hormone Releasing Hormone [Nal-Glu] Antagonist, Peptide (#MBS659559); My Biosource, San Diego, CA, USA) (Fig. 1). The dosage ranged from 5.0 mg/kg to 1.25 mg/kg throughout the protocol as the animal body weights increased. Fifteen animals from each group were sacrificed immediately after the injection protocol at Day 50 (C, GnRH-a). The remaining animals recovered from the saline or GnRH-a injections for 135 days and a subset of each group was sacrificed at Day 185 ((C-R) (n = 15) and (G-R) (n = 15)). The remaining animals had an ovariectomy surgery (OVX) at 185 days of age and were sacrificed 40 days later (C-OVX) (n = 15) and (G-OVX) (n = 15). After sacrifice (50, 185 and 225 days) (n = 15∕group), femurs and tibia were mechanically tested and femurs were scanned using micro CT and serum C-terminal telopeptides (CTX) and insulin-like growth factor 1 (IGF-1) were measured.

All animals were given food and water ad libitum. All animals were monitored daily for vaginal opening, an indicator of pubertal onset (Ojeda et al., 1976). Body weights were taken daily until the onset of puberty and then every 4 days. Growth rates were calculated (grams/week) for early puberty ((weight day 37-weight day 23)/2) and late puberty ((weight day 50-weight day 37)/2); puberty typically occurs at day 30. On the day of sacrifice, body weight was measured. Animals were anesthetized using CO2 (Toft et al., 2006) and blood was taken via a cardiac puncture. The animals were then killed by an overdose of CO2. Uterus, ovaries, retroperitoneal and gonadal fat pads, and triceps surae muscle tissue were collected and weighed. Femora and tibiae were removed and cleaned of soft tissue. The right femora and tibiae were mechanically tested and ashed for bone mineral content while the left femora were fixed in 10% buffered formalin for 48 h to one week for micro-CT imaging. All procedures were approved by the Institutional Animal Care and Use Committee (IACUC) at Temple University (3396). Gonadotropin-releasing hormone antagonists (GnRH-a) injections have successfully delayed the onset of puberty by hypothalamic suppression resulting in low estrogen levels in female rats and have the advantage that normal hypothalamic-pituitary function is restored after cessation of injections (Roth et al., 2000).

Figure 1 Timeline of GnRH-antagonist injection protocol, recover and ovariectomy surgery.

Mechanical testing

Mechanical strength of the right femora and tibiae was measured under three-point bending using a materials testing machine (Bose Electroforce, Eden Prairie, MN, USA) containing a 450 N load cell. Bones were placed on the lower supports anterior side down and loaded in the anterior–posterior plane. Span length of the lower supports was maximized to minimize the effect of shear loading. Span lengths were as follows for each time point: injection protocol (femur-16.8 mm, tibia-22.3 mm), recovery (femur-21.7 mm, tibia-26.3 mm), post-OVX (femur-20.2 mm, tibia-25.2 mm). Prior to testing the bones were thawed in saline to ensure hydration and then loaded to failure at a rate of 0.05 mm/s. During testing, force and displacement data were collected. Bending moment was calculated from the force data (M = FL∕4) (N mm) and displacement data were divided by (L2∕12) (mm∕mm2) where L is the distance between the lower supports (span length). Mechanical properties of the whole bone were then determined from the moment vs. normalized displacement curves. Mechanical properties included: peak moment (N mm) (the maximal load the specimen sustained) and stiffness (N mm2) (slope of the initial linear portion of the moment-displacement curve). Both the peak moment and the stiffness are measures of bone structural strength and stiffness and both measurements are influenced by the bone material/composition measured by ash fraction and the distribution of that material (bone geometry) measured by polar moment of inertia.

Micro-CT analysis

The left femora were fixed in 10% buffered formalin for 48 h to one week, stored in 70% ethanol, and scanned in an ex vivo µCT scanner (SkyScan 1172; SkyScan, Aartslaar, Belgium) to measure both geometrical changes in the cortical bone sites and structural changes in trabecular and cortical bone sites. The Skyscan-1172 has a sealed micro focus X-ray tube that emits from 20 keV to 100 keV energy with a 10 megapixel (4,000 × 2,096), 12-bit cooled CCD camera. Scanning was performed using a source voltage of 60 kV and source current of 167 µA with a 0.5 mm Al filter to minimize the beam hardening from the polychromatic nature of the sealed X-ray source. Scans were done with a rotation step of 0.4°through 180°and a pixel size of 7.7 µm. The Feldkamp cone-beam reconstruction algorithm was used to reconstruct the three-dimensional cross sections along with addressing the ring artifact reduction of 10 and beam hardening corrections of 20%. Approximately 400 slices in both the metaphyseal trabecular and mid-diaphyseal cortical regions were analyzed. Region of interest for the trabecular bone in the distal femora were positioned at one image distal to the point that the growth plate disappeared to quantify metaphyseal trabecular bone. Cortical region of interest was taken from the midshaft of the femur (200 slices above and below). Trabecular bone analysis included percent bone volume (BV/TV) (volume of bone relative to the total volume of the region of interest), trabecular thickness (Tb.Th), trabecular number (Tb.N), trabecular separation (Tb.S), while cortical bone analysis included total cross-sectional area (T.Ar), cortical bone area (Ct. Ar), marrow area (Ma.Ar), cortical thickness (Ct.Th), cortical volume fraction (BV/TVcort), polar moment of inertia (J). All parameters were calculated according to the ASBMR standards (Parfitt et al., 1987; Bouxsein et al., 2010).

Bone mineral content

After mechanical testing, the right femora were flushed with phosphate-buffered saline to discard the marrow. Dry weight of the bones was determined after drying in an oven at 100 °C for 24 h. Ash weight was determined after ashing the bone in a muffle furnace (Fischer Scientific, Hampton, NH, USA) at 700 °C for 24 h. Ash fraction was calculated as ash weight/dry weight.

Blood chemistry

C-terminal telopeptides or carboxy-terminal collagen crosslinks (CTX) were measured in serum using an immunoenzymometric assay (Rat-Laps EIA; Immunodiagnostic Systems Inc., Fountain Hills, AZ, USA). The detection limit of the assay was 2.0 ng/mL. Serum insulin-like growth factor 1 (IGF-1) was measured using an immunoenzymometric assay (Rat/Mouse IGF-1; Immunodiagnostic Systems Inc., Fountain Hills, AZ, USA). The sensitivity of the assay was 63 ng/mL.

Data analysis

Two-way ANOVA (2 groups (GnRH-a and control) X 3 time points (Injection Protocol, Recovery, post-OVX)) was computed to determine interactions between the GnRH-a and control groups for the dependent variables. A Sidak’s multiple comparison test was used to determine significant interactions between groups and time points. A level of p ≤ 0.05 was considered significant. The relationship of body weight to polar moment of inertia and cortical area were determined by regression analysis (p < 0.05). The relationship of polar moment of inertia to cortical area was determined by regression analysis (p < 0.05). All statistical analyses were performed in GraphPad (GraphPad Prism version 6.00 for Windows; GraphPad Software, San Diego, California, USA). Mechanical variables were normalized with a linear regression-based correction using body weight (Di Masso et al., 1997) since at sacrifice there was a significant difference in body weight between the groups (non-normalized values are also presented). All variables with an R2 level greater than 0 were normalized to avoid choosing an arbitrary R2 value as a cut-off for normalization.

Results

Vaginal opening-delayed puberty

A significant delay in vaginal opening (an indicator of the onset of puberty) was evident following the 25-day GnRH-a injection protocol in all 3 GnRH-a groups (GnRH-a, G-R, G-OVX) (Fig. 2). Only 40–60% of the GnRH-a groups had a vaginal opening by day 50, the end of the injection protocol (Fig. 2). The average day for VO in the control groups (C, C-R, C-OVX) was 34 days of age. The G-R and G-OVX groups had 100% of the group reach puberty, indicated by a vaginal opening, by 60 days of age, almost 2 × later than the control groups.

Figure 2 Comparison of the timing of Vaginal Opening (VO) of all groups (C, GnRH-a, C-R, G-R, C-OVX, G-OVX).

The cumulative percent of animals with VO is displayed per age (days). The GnRH-a, G-R, G-OVX groups have a lower percentage of animals with VO at later ages. The C, C-R and C-OVX groups reached 100% VO by day 39.

Anthropometrics

Uterine weights were 72% lower in the GnRH-a group compared to control a further indication of the success of the GnRH-a injection protocol (Fig. 3A). Uterine weight recovered during the recovery phase, there was no difference between the C-R and G-R groups (Fig. 3A). Uterine weights were approximately 60% lower following the OVX surgery in both groups; C-OVX and G-OVX (Fig. 3A). The hypothalamic suppression via GnRH-a injections significantly increased the growth rates of the GnRH-a groups from week 5 to 7 (late puberty) compared to control (Fig. 3B). Growth rates in the control groups significantly slowed down weeks 5–7 (late puberty) compared to weeks 3–5 (early puberty) (Fig. 3B). There was a significant group effect for body weight at sacrifice (p = 0.0015). The GnRH-a groups were significantly heavier than controls. Body weights increased at each time point, recovery and post-OVX (Fig. 3C).

Figure 3 Uterine Weights, Growth Rates and Body Weights for the three time points, Injection Protocol, Recovery, post-OVX.

(A) Uterine Weights (g) for C (black bars) and GnRH-a (white bars) groups, (B) Growth Rate (g/week) for Early Puberty was calculated ((weight day 37-weight day 23)/2), Late Puberty was calculated ((weight day 50-weight day 37)/2). (C) Body Weight (g) at sacrifice for C (black bars) and GnRH-a (white bars) groups. ∗ indicates significant difference from control values at same time point or from early puberty; # indicates significant difference from Injection Protocol or from Control; γ indicates significant difference from Recovery.

Blood chemistry

There was a significant interaction between group and time point for CTX, a marker of bone resorption, indicating that differences between groups occurred at different time points. CTX levels for the GnRH-a group were significantly higher (59%) compared to control following the 25-day injection protocol (Fig. 4A). Simple main effects of time point within controls was significant, the CTX levels were the lowest at the recovery time point with no differences between acute and post OVX levels. However the CTX levels for the GnRH-a group were the highest during the acute treatment phase even higher than levels post-OVX (Fig. 4A). A significant interaction was also found for IGF-1 serum levels. No differences were found between GnRH-a and C following the injection protocol or recovery period, however IGF-1 levels were 33.3% lower in the GnRH-a group between the injection protocol and the recovery stage (Fig. 4B).

Figure 4 Serum markers of bone resorption (CTX) and formation (IGF-1).

(A) serum CTX (ng/mL) levels (B) serum IGF-1 (ng/mL) levels (n = 8--10). ∗ indicates significant difference from control values at same time point. # indicates significant difference from Injection Protocol; γ indicates significant difference from Recovery.

Table 1 Bone strength and structural parameters and body composition.

Whole bone mechanical parameters measured from three-point bending of the femoral and tibial diaphysis, geometry and structural measurements from micro CT analysis, bone and body composition measures. Mean (SD).

	Injection protocol	Recovery	Post-OVX	2-way ANOVA p-values	
Mechanical properties	
Femur-peak moment (N mm)	Control	203 ± 31	557 ± 140**,*	797 ± 95**,***	G×T: p=0.0214	
	GnRH-a	231 ± 19	713 ± 130**	823 ± 90**,***		
Femur-stiffness (N mm2)	Control	9,880 ± 2,010	42,920 ± 10,880**	73,360 ± 9,490**,***	G: p=0.0388	
	GnRH-a	9,920 ± 1,240	51,400 ± 13,050**	77,010 ± 8,680**,***	T: p<0.0001	
Tibia-peak moment (N mm)	Control	189 ± 17	528 ± 83**	575 ± 96**,***	T : p<0.0001	
	GnRH-a	197 ± 18	563 ± 109**	621 ± 92**,***		
Tibia-stiffness (N mm2)	Control	11,003 ± 1,497	50,580 ± 11,488**	57,810 ± 13,240**,***	T: p<0.0001	
	GnRH-a	11,650 ± 1,140	56,420 ± 12,980**	62,070 ± 13,040**,***		
Geometry	
J (mm4)	Control	8.7 ± 1.3	13.1 ± 1.8**	15.8 ± 2.3**,***	G: p=0.0284	
	GnRH-a	9.3 ± 1.0	15.6 ± 2.7**	16.0 ± 2.0**,***	T: p<0.0001	
Mass and structure	
Ct.Ar (mm2)	Control	3.5 ± 0.3	6.2 ± 0.4**	6.8 ± 0.4**,***	T: p<0.0001	
	GnRH-a	3.8 ± 0.2	6.4 ± 0.7**	6.8 ± 0.5**,***	G: p = 0.0619	
T.Ar (mm2)	Control	9.3 ± 0.9	9.4 ± 0.8	10.4 ± 1.0**	T: p<0.0001	
	GnRH-a	9.3 ± 0.8	10.4 ± 0.9	10.4 ± 0.7**		
Ma.Ar (mm2)	Control	5.8 ± 0.9	3.2 ± 0.5**	3.7 ± 1.1**	T: p<0.0001	
	GnRH-a	5.5 ± 0.8	4.0 ± 0.5**	3.6 ± 0.5**		
Ct.Th (mm)	Control	.22 ± .03	.52 ± .12**	.51 ± .12**	T: p<0.0001	
	GnRH-a	.22 ± .03	.53 ± .11**	.60 ± .05**		
Bone mineral content	
Ash fraction %	Control	58 ± 4	67 ± 1	65 ± 2	T: p<0.0001	
	GnRH-a	60 ± 2	66 ± 2	65 ± 5		
Body composition	
Triceps Surae muscle Mass/BW (%)	Control	0.65 ± 0.04	0.67 ± 0.06	0.63 ± 0.05***	G×T: p = 0.0455	
	GnRH-a	0.62 ± 0.03	0.68 ± 0.04**	0.66 ± 0.05**		
Total fat pads (g)	Control	1.2 ± 0.6	10.0 ± 4.3**	12.6 ± 3.6**	T: p < 0.0001	
	GnRH-a	2.0 ± 0.7	12.4 ± 2.7**	13.8 ± 4.3**	G: p= 0.0640	
Notes.

Values are presented as mean + SD.

* Indicates significant difference from control values at same time point.

** Indicates significant difference from Injection Protocol.

*** Indicates significant difference from Recovery.

Mechanical properties

There was a significant interaction between group and time point for peak moment of the femur. The G-R group was 28% stronger following recovery compared to C-R (Table 1). However, no differences in peak moment between GnRH-a and C groups were found following the injection protocol or post-OVX (Table 1). Stiffness in the femur was significantly greater in the GnRH-a groups compared to control. No group differences in mechanical strength were found in the tibia (Table 1).

Micro-CT analysis-cortical

The polar moment of inertia (J) of the femur was significantly larger in the GnRH-a groups compared to control. In addition, cortical area, the amount of bone, trended higher (p = 0.0619) in the GnRH-a groups. The total area, marrow area and cortical thickness were not significantly different between groups. Polar moment of inertia increased with later time points compared to the first time point following the injection protocol (Table 1). The cortical area did significantly increase with each time point with the greatest cortical area being post OVX in both groups (p < 0.0001). Marrow area (Ma.Ar) decreased significantly in both groups from the post injection protocol time point to the recovery but remained constant from recovery to post OVX (p < 0.0001). Total area (T.Ar) was greater post OVX compared to the acute time point but no differences were detected between the acute and recovery time points. Cortical thickness was significantly thinner at the acute time point compared to recovery and post OVX.

Micro-CT analysis-trabecular

Both trabecular bone volume and structure were acutely affected by the GnRH-a injection protocol and these deficits lasted through maturity and post-OVX. GnRH-a group had less percent bone volume (BV/TV) at each time point compared to control. Following the injection protocol, BV/TV was 31% lower in the GnRH-a group compared to C, a similar deficit in BV/TV was also measured following recovery and post-OVX. The simple main effect of time point within each group (GnRH-a and Control) were significant, the greatest BV/TV for both groups was post recovery; the two other time points were less but post-OVX was greater than the BV/TV following the injection protocol (Fig. 5A). The greater BV/TV in the control groups was due to both a larger trabecular number and a thicker trabecula. The trabecular number was greater in the control group compared to GnRH-a but both groups had the greatest number of trabecula at recovery and lower numbers post-OVX (Fig. 5B). Following the injection protocol the control group had a third greater number of trabeculae, the larger number continued at recovery and post-OVX. The trabeculae were also thicker in the control groups compared to GnRH-a groups. The trabeculae were thickest at recovery and thinnest following the injection protocol (Fig. 5C). On average, the trabeculae of the control groups were 3% thicker than the GnRH-a groups. The lowest trabecular separation was measured at recovery but after the OVX surgery the trabecular separation increased (Fig. 5D).

Figure 5 Structural measurements from micro CT analysis.

(A) Percent trabecular bone volume (BV/TV) (%), (B) Trabecular Number (Tb.N) (1/mm), (C) Trabecular Thickness (Tb.Th) (mm), (D) Trabecular Separation (Tb.Sp) (mm) (n = 9--15). ∗ indicates significant difference from control values; # indicates significant difference from Injection Protocol; γ indicates significant difference from Recovery.

Bone mineral content

The ash fraction was significantly less following the injection protocol compared to recovery and post-OVX time points in both groups. No difference in ash fraction was found between the C and GnRH-a groups.

Body composition

Triceps Surae muscle mass per body weight (%) was significantly different depending on group (interaction). However, post-hoc tests did not indicate difference between groups at any of the time points (Table 1). The total weight for the retroperitoneal and gonadal fat pads were trending larger in the GnRH-a groups (p = 0.064). Total fat pad weights were significantly higher than the injection protocol in both the control and GnRH-a groups at recovery and post-OVX (Table 1).

Efficiency of bone structure

Both groups (Control and GnRH-a) built competent structures during the injection protocol, during recover and post-OVX. The polar moment of inertia increased linearly with the increases in body weight and no differences in slope or intercepts were detected between groups (Fig. 6A). In addition, cortical area (Ct.Ar) also increased with body weight in a similar manner for both groups (Fig. 6B). The efficiency of the bone structure expressed by the regression of cortical area and polar moment of inertia indicated that as the cortical area increased so did the polar moment of inertia. The larger polar moment of inertia mechanical strength of the femoral diaphysis increased. The relationship between cortical area and polar moment of inertia were not different between groups (Fig. 6C).

Figure 6 Linear regression of body weight and bone structural variables for the GnRH-a and control groups.

(A) There was a significant relationship between body weight and polar moment of inertia but no difference between groups (C: R2 = .9188; GnRH-a: R2 = .7846). (B) There was a significant relationship between body weight and cortical area but no difference between groups (C: R2 = .8909; GnRH-a: R2 = .9079). (C) There was a significant relationship between polar moment of inertia and cortical area but no difference between groups (C: R2 = .7893; GnRH-a: R2 = .9043).

Discussion

Previously we reported a short term deficit in bone strength following hypothalamic suppression during growth (Yingling & Khaneja, 2006; Yingling & Taylor, 2008) with cortical bone strength recovering by 6 months of age (Yingling & Khaneja, 2006). In the current study, bone structure and strength were assessed at maturity following ovariectomy (OVX) surgery while still maintaining the acute hypothalamic suppression via gonadotropin releasing hormone antagonist injections during growth; thus, investigating the response of both cortical and trabecular bone growth on bone strength, geometry and mineral at maturity, particularly after the menopause. The hypothesis that environment during bone development affects the structure at maturity was partially supported by our data. Functional bone strength was maintained through architectural adaptations in the cortical bone envelope throughout the lifespan; however, a lower trabecular bone mass during growth was retained through recovery and post ovariectomy surgery. Our data support the hypothesis that trabecular bone is more vulnerable to factors negatively affecting growth and are less likely to recover from these deficits.

An acute deficit in bone strength (cortical diaphysis) was not found the in current study following a delayed pubertal onset, hypothalamic suppression. In fact, following recovery the femoral peak moment was greater in the GnRH-a group (G-R) compared to control (C-R). The cortical bone architecture specifically the polar moment of inertia was greater in the GnRH-a groups which rescued bone strength following the delayed puberty. Cortical bone stiffness increased with aging (recovery and post-OVX). However, trabecular bone volume (BV/TV) was significantly lower in the GnRH-a groups compared to control immediately following the injection protocol, after recovery and post OVX surgery. Both trabecular number and thickness were lower for all time points as well.

GnRH-a injections successfully suppressed hypothalamic function

The GnRH-a injections effectively suppressed hypothalamic function as indicated by the delay in the age of vaginal opening, an indicator of puberty in the rat. In addition, uterine weights in the GnRH-a groups where on average 72% lower compared to control groups. Suppression of the hypothalamus lowers GnRH secretion which in turn suppresses luteinizing hormone (LH) and follicle stimulating hormone (FSH) and thus suppresses estrogen secretion from the ovaries and lowers serum estrogen levels (Schally, 1970). Lower uterine weights are indicative of suppressed estrogen release from the ovaries. Although estrogen levels were not measured in the current study, previous data corroborates suppressed serum estradiol levels using the GnRH-a injection protocol in pre-pubescent female rats (Yingling & Khaneja, 2006; Yingling et al., 2007; Yingling & Taylor, 2008; Saine et al., 2011). In addition, serum CTX levels were significantly increased by 59% in the GnRH-a group an indicator of increased bone resorption also associated with suppressed estrogen secretion.

Trabecular bone volume significantly lower in the GnRH-a groups

The increase in bone resorption indicated by increased CTX levels accompanied a 31% lower trabecular bone volume (BV/TV) in the GnRH-a groups compared to control immediately following the injection protocol. The deficit in trabecular bone volume was not improved during the recovery period and following OVX surgery, a 34% deficit in trabecular BV/TV remained. We have previously reported a lower yet not statistically significant trabecular volume via histomorphometry using a similar GnRH-a dosage (Yingling et al., 2007). Using texture analysis, lower BV/TV at specific orientations was found and remained lower at maturity, specifically in orientations that do not resist loading (Yingling et al., 2007). Long-term suppression of trabecular bone was also reported by Rakover et al. (2000) using peripheral quantitative computerized tomography (pQCT) and dual energy X-ray absorptiometry (DXA) (Rakover et al., 2000). Trabecular bone catch-up growth was not evident in the GnRH-a group, however, the changes in trabecular BV/TV over time were similar between groups. These data suggest that following a transient delay in pubertal onset, the growth patterns remain similar in both GnRH-a and control animals with no restoration of trabecular bone volume in the distal femur. Similar failures to regain bone mass have been reported in elite female athletes following delayed puberty and amenorrhea during young adulthood (Warren et al., 2002; Warren et al., 2003), specifically amenorrheic dancers receiving hormone replacement therapy for 2 years had no increase in bone mineral density (BMD) compared to placebo or control groups.

Reduced trabecular bone volume was primarily due to fewer trabeculae following the injection protocol (29% less); the thickness of the trabeculae was only 3–5% thinner in the GnRH-a groups. In osteoporotic women, trabecular bone is typically reduced due to trabecular thinning and loss of trabecular connectivity (Aaron, Makins & Sagreiya, 1987; Borer, 2005). A previous study using the GnRH-a protocol at a 100 µg/day dosage measured a loss in percent trabecular bone volume (BV/TV) due to a larger trabecular separation and lower trabecular number (Yingling et al., 2007). The current data are the first to suppress trabecular bone during growth, and then add an OVX protocol at maturity; the result was a lower trabecular number and a continued thinning of the trabeculae throughout the life span. Further investigation is needed to determine if this pattern would be replicated in the vertebral spine and other trabecular bone regions.

Cortical bone strength preserved following GnRH-a Injections

Interestingly while acquisition of trabecular bone volume was reduced, there were no differences in cortical bone area (p = 0.06), total area or marrow area, only the polar moment of inertia differed significantly between groups. Cortical bone adapted to the acute GnRH-a injection protocol during development by varying the architecture within the diaphysis to maintain a sufficiently strong bone. Cortical bone strength is maintained throughout the aging process by a coordination of morphological and compositional traits including cortical area, geometry and tissue mineralization. These traits can co-vary in order to build a functional bone structure (Jepsen et al., 2013; Schlecht & Jepsen, 2013). The GnRH-a injections did not affect mineralization as measured by ash fraction in the current study. The bone mass measured by cortical area was trending higher in the GnRH-a groups yet was correlated with the increased body weight (Fig. 6B). Bone geometry measured by polar moment of inertia (distribution of bone mass in the cortical diaphysis) was greater resulting in greater femoral stiffness in the GnRH-a groups due to increase in bone distributed away from the bending axis. The cortical diaphysis was able to maintain optimal structure during the period of hypothalamic suppression and even increase strength during recovery due to the larger polar moment of inertia.

Body weight is commonly increased following GnRH-a injections (Rakover et al., 2000; Roth et al., 2000; Yingling & Taylor, 2008) and is due to increased growth rates during late puberty. Increased growth rates have been associated with strength deficits (Rawlinson et al., 2009) potentially due to increased woven bone growth on the periosteal surface. Although bone formation and woven bone were not measured in this study there were no changes in ash fraction, a measure of mineralized bone. Both cortical area and polar moment of inertia correlated with the larger body weights in the GnRH-a groups indicating that the change in body size was matched with appropriate bone structural changes in the current study. The cortical area was also correlated with the polar moment of inertia an indication of structural efficiency, as bone mass increases so should the distribution of that mass in order to increase bone strength. All correlations between bone variables and body weight were similar between the control and GnRH-a groups.

The dosage of the GnRH-a injections and timing of injections may affect the short term effects on the cortical bone strength. One explanation for the maintenance of cortical bone strength following the injection protocol may be moderate estrogen suppression in the current study. Approximately 50% of all animals in the GnRH-a group reached puberty by day 50, the end of the injection protocol. Previous studies using a similar dose of GnRH-a reported lower estrogen levels (27%) following a GnRH-a injection protocol (Rakover et al., 2000; Yingling & Khaneja, 2006; Yingling & Taylor, 2008) but the subset of animals that never reached pubertal onset had a greater suppression of estrogen (50%) indicating that complete delay of puberty reduces estrogen to a larger extent and may have greater effects on cortical bone structure (Yingling & Khaneja, 2006).

These data in the current study suggest a difference in response to hypothalamic suppression (GnRH-a injections) between cortical and trabecular bone. These data suggest that even as serum bone resorption levels increased, cortical bone structure was altered to preserve bone strength and function. The difference in cortical and trabecular bone sensitivity to estrogen suppression has also been reported in human studies of women with oophorectomy (Genant et al., 1982) who needed a 4-fold higher dose of estrogen to prevent trabecular bone loss compared to cortical bone loss. One theory is that trabecular bone has more ERβ (Bord et al., 2001; Khosla, 2008).

Limitations of the GnRH-a injection model

Multiple variables will affect skeletal development during a delay in puberty including the age at which the delay occurs, the duration of delay and the severity of the delay. To understand fully the mechanisms that affect mechanical properties due to a delay in pubertal development and the long term consequences of such a delay necessitates an animal model that reflects closely the clinical characteristics of delayed puberty. The GnRH-a model does replicate key factors reported clinically in patients with delayed puberty. Uterine and ovarian weights were significantly suppressed suggesting that estradiol levels are suppressed in animals using this model similar to human athletes with primary and secondary amenorrhea (Drinkwater et al., 1984; Warren et al., 1991; Pettersson et al., 1999). However, there are limitations to this model. The increased body weights are not consistent with clinical populations. The GnRH-antagonist model of delayed pubertal development only affects the hypothalamic-pituitary-gonadal axis and clinically the condition of delayed pubertal onset may be a more complex interaction of somatic and reproductive maturation. Studies that treated amenorrheic dancers for 2 years with hormone replacement therapy found no difference in BMD between treated and placebo groups (Cumming, 1996; Warren et al., 2003) suggesting that estradiol is not the only factor in the dancer’s bone loss.

Conclusion

Following a delay in pubertal onset during development, functional bone strength was maintained through architectural adaptations in the cortical bone envelope throughout the lifespan however, a lower trabecular bone mass during growth was retained through recovery and post ovariectomy surgery. An increase in the polar moment of inertia offset increased bone resorption. Cortical bone adapted to an acute environmental insult by varying the architecture within the diaphysis to maintain a sufficiently strong bone. The acquisition of trabecular bone volume was reduced after the injection protocol that delayed pubertal onset trabecular bone volume was not restored following recovery and remained lower post OVX surgery. The data suggest that the trabecular bone was more sensitive to changes in the systemic effects resulting from hypothalamic suppression and bone volume was not restored following recovery.

Supplemental Information

Supplemental Information 1 Data set

Click here for additional data file.

We thank the members of the Skeletal Adaptation and Development laboratory in the Department of Kinesiology at Temple University.

Additional Information and Declarations

Competing Interests

Author Contributions

Animal Ethics

Data Availability

The authors declare there are no competing interests.

Vanessa R. Yingling conceived and designed the experiments, performed the experiments, analyzed the data, contributed reagents/materials/analysis tools, wrote the paper, prepared figures and/or tables, reviewed drafts of the paper.

Kathryn A. Mitchell performed the experiments, analyzed the data, wrote the paper.

Megan Lunny performed the experiments, analyzed the data.

The following information was supplied relating to ethical approvals (i.e., approving body and any reference numbers):

Institutional Animal Care and Use Committee (IACUC) at Temple University (3396).

The following information was supplied regarding data availability:

The research in this article did not generate any raw data.

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
