# Peer review of "Acute hypothalamic suppression significantly affects trabecular bone but not cortical bone following recovery and ovariectomy surgery in a rat model"

_PeerJ, doi:10.7717/peerj.1575_

## Round 0.1 · original submission · Major Revisions

· Academic Editor

Major Revisions

Both reviewers provide detailed comments requesting clarification of various points. Please address these in your revised manuscript or rebuttal letter.

In addition to the reviewer comments, please check the methodological details and clarify the contradictions in the results section as pointed out below.

Methods: Which GnRH antagonist was injected? The calculation of early and late puberty is not clear. Did you start measurements in week 3 or with 25 days? Is the 5th week included in early or in late puberty? Day 30 is not equal to week 4. The bending moment is given in Nmm in line 119 and in N/mm in table 1. Line 133, the unit 60KeV/167 µA-1 seems wrong. Did you analyse the complete triceps surae muscle mass (line 263) or only the gastrocnemius muscle (table 1)? Please use the signs of significance consistently throughout figure 3, 4 and 5.

In the results section you will need to clarify whether stiffness in the femur was greater in the control groups (line 219) or lower (table 1). The significant higher ash fraction in the GnRH-a group (line 258) does not correspond to table 1. The time dependent changes in ash fraction (table 1) are in contradiction to the abstract. The vaginal opening in the first GnRH-a group was around 40% at day 50 according to figure 2, line 182 states 50-60%.

Reviewer 1 ·

Basic reporting

This reviewer is not sure that the submission clearly defines the research question. This stems from the fact that the ‘gap in our knowledge’ is not clearly identified and, therefore, conclusions regarding how this study contributes to filling that gap are unfocussed. For instance the final paragraph of the introduction fails to identify what purpose is served by studying the OVX response.

Experimental design

Methods are probably described with sufficient information to be reproducible by another investigator but are nevertheless somewhat flawed. Thus:
1. The authors injected 0.24mg per dose to each rat and did not report how this relates to the body weight. Previous publications from Yingling (Joshi et al., 2011 and Saine et al., 2013) reported 2.5 mg/kg GnRH injection. Injections were performed for 25 days and so the authors need to clarify injection regime with respect to animal weights.
2. Was estradiol levels measured and if so the authors need to provide data for all groups. Uterine weight and vaginal opening are good indications, however, referencing previous publication for estradiol reduction is not methodologically sufficient.
3. The authors analyse 400 slices in both metaphyseal trabecular and mid-diaphyseal cortical region. The landmark used to select trabecular bone needs clarification. Furthermore, the authors do not seem to have corrected trabecular and cortical analysis for bone length. As bones used in analysis are from rats from various time points, the analysis need to be repeated considering a percentage (for example 5% of total length) or the existing data be corrected for bone length (is bone length modified?).
4. The main flaw is in the experimental design which fails to include a group of rats which were treated identically through to recovery, but then are retained for a further 40days without OVX. This is essential if one wishes to interpret the relative impact of OVX; the data suggest no differences in the response of the two groups to OVX but one can’t be sure without appropriate control animals.

Validity of the findings

The main conclusion from this manuscript on the sensitivity of trabecular bone is not fully justified by the stated method of analysis of trabecular bone.
The authors describe the effect of GnRH treatment on cortical bone and report cortical area. The authors also need to report cortical thickness as another measure of bone mass and geometry.
Data from cortical analysis indicate redistribution of bone and so the authors need to report the effect of GnRH on entire bone or select multiple cortical regions for analysis– the selection of the mid-shaft alone might not provide the whole picture.
The description in the results section does not match the Figures 3A-C

·

Basic reporting

Excellent.

Experimental design

Excellent.

Validity of the findings

Excellent.

Additional comments

This is a strong piece of research and generally well-presented manuscript that shows the effect throughout life of pubertal delay on bones in females, using a rat model of pubertal delay. A major strength of the paper is that it makes a thorough investigation of bone physiology – from structure as assessed with micro CT, to bone turnover marker and IGF-1 levels in the circulation, to bone strength, all at key time points throughout life, notably post-OVX as a model of menopause. Some comments below could help to further clarify and improve this manuscript.

MAJOR

1. The Abstract needs greater clarity about how many rats were in each group at the outset and at each time point, as the numbers in the Abstract do not match the numbers given clearly in the Methods section. The Abstract also needs to state explicitly that the GnRH-a treatment was only for 25 days.

2. The Abstract should provide a short phrase to explain what ‘moment of inertia’ is a measure of (strength? Or only architecture?). Also, is this different from ‘polar moment of inertia’ and ‘peak moment’? If not, the same term should be used throughout the manuscript.

3. The manuscript should provide some phrases throughout to briefly tell the reader what the following measures are an indication of: ‘moment of inertia’, ‘polar moment of inertia’ and ‘peak moment’. Are these terms one and the same (in which case a single term should be used)? How do they differ from each other, or relate to each other? How do these measures relate to the results from the bone breaking experiments? Readers who are interested in this paper may come from the human bone field, and may not be familiar with such measures in rodents.

4. The manuscript should mention whether the findings may or may not be relevant to children who are under treatment with GnRH-a to delay the onset of puberty, in response to signs of early puberty onset. For example, is there any evidence that children who have been treated with GnRH-a have reduced bone mass or increased fracture rates in adulthood compared to children that have not been treated as such? Or would an effect such as that seen in this manuscript be expected only if puberty in children were delayed beyond the normal age of puberty, as in very thin athletes or dancers, for example?

5. Lines 35-37 should provide more information about the direction of this relationship between age of menarche and stress fractures.

6. Lines 59-62 allude to ‘few studies’ that have investigated parameters that are investigated in the current manuscript, but this implies that other similar studies have been performed. The manuscript should clearly state what those other studies have shown, and how this study is different and how it will fill gaps in knowledge.

7. Lines 101-103: Was ethics approval really granted to perform cardiac puncture on the rats prior to killing them? There is no mention of anaesthesia used during this process of cardiac puncture prior to CO2 overdose.

8. Line 257 and elsewhere in the manuscript – it will be helpful to include a brief phrase to explain what ash fraction is an indicator of. Bone mineral content?

9. The Discussion section is very long (8 pages) and as such the main message is diluted with an excess of text, much of which reads like a repeat of the Results section and makes some reference to figures (Line 330). The A shorter Discussion would provide a stronger take-home message.

10. Line 407 mentions an increase in circulating IGF-1 levels as a possible explanation for the current findings, but there is no evidence of increased IGF-1 levels between the treated and control groups in the current study.

11. The Conclusion from Line 461 should use more complete sentences in order to be more self-explanatory, to benefit the readers who will skip straight to this section rather than reading the whole Discussion. At present, the Conclusion would not be understood by someone who had not read the whle paper or Discussion.

12. It seems incorrect to use line graphs in Figure 5, because the points are not the same animals. A bar graph would likely be more appropriate.

MINOR

13. The Abstract has a word missing after the words ‘recovered from’.

14. Lines 91-94 in the Methods section repeats information given earlier in that section, and should therefore be deleted or merged with the earlier text on this topic.

15. The table would benefit from reducing the number of significant figures on the numbers. For example, 77014.4 would be clearer and easier to read and compare if it were ‘summarised’ to 77010.

---

## Round 0.2 · Minor Revisions

· Academic Editor

Minor Revisions

There are still some discrepancies that need clarification. In addition to the points mentioned by the reviewer below, could you please adress the following points:

Abstract
"GnRH-a injections suppressed uterine weights (74%)" – compared to - 72% in Fig. 3 and line 202 and 325
"Bone stiffness was similar in C and GnRH-a groups" – compared to - significantly greater in the femur in line 232 and table 1
"Ash content and cortical bone area were similar at all time points" – compared to - time dependent changes in table 1 and line 271

Methods and Results
Please include the name of the GnRH antagonist as given in the responding letter.
Include a sentence about cortical thickness (Ct.Th.) in methods (line 156) and results (line 247). The abbreviation is only presented in table 1 and not explained in the manuscript, but should be included (see reviewer comment to the first version).
The meaning of BV/TV is not quite clear. Could you explain the abbreviation at first mention. Shouldn’t it be Trabecular volume (TV) / total bone volume (BV) = TV/BV?

Discussion
Line 311 - Is cortical diaphysis identical to or represented by Ct.Th. in table 1? Could you add this information.
Line 439 ...increased IGF-1 levels – These are not shown. There is no difference in Fig. 4B. Should there be a significance? See the comment below.
Figure 4B
According to your legend a difference between injection protocol and recovery should be labelled with #. If there is an (additional) difference between GnRH-a and control at both time points please place the * above the GnRH-a column as in Fig. 4A
Figure 5
It is a bit confusing that the labelling colors are switched. Control is black in Fig. 2, 3, 4, and 5 A and B. However, control is white in Fig. 5 C and D and in Fig. 6.

·

Basic reporting

Excellent.

Experimental design

Excellent.

Validity of the findings

Excellent.

Additional comments

This manuscript is much improved. Perhaps a reference should be given to support the anesthetic effect of CO2. My points numbers 13 and 15 in my original review do not seem to have been adjusted (there are still too many significant figures in the table for clarity). In addition, there are typographical errors in the manuscript (e.g. on page 7 (it should be ‘femora’ not ‘femura’) and others) that should be proofread out of the paper. On page 13, when bone mineral content is mentioned, the authors should say whether or not there is any difference between groups in ash fraction. Page 21 still mentions an increase in serum IGF-1 with GnRH-a treatment in rodents, but this was not observed in the current m/s, so this text needs modification to reflect that.

---

## Round 0.3 · accepted · Accept

· Academic Editor

Accept

The manuscript has been greatly improved by the revisions.